# Increased Knowledge of Adult-Onset Dystonia Amongst Medical Students via Brief Video Education: A Systematic Review and Cohort Study

**DOI:** 10.3390/geriatrics7030058

**Published:** 2022-05-18

**Authors:** Sana Khan, Nina Sowemimo, Jane Alty, Jeremy Cosgrove

**Affiliations:** 1School of Medicine, University of Leeds, Worsley Building, Leeds LS2 9JT, UK; sana.khan8@nhs.net (S.K.); nina.sowemimo@nhs.net (N.S.); 2Department of Neurology, Great George Street, Leeds Teaching Hospitals NHS Trust, Leeds LS1 3EX, UK; jane.alty@nhs.net; 3Wicking Dementia Research Centre, University of Tasmania, Hobart 7001, Australia

**Keywords:** dystonia, medical education, curriculum, neurology, online

## Abstract

Most doctors have limited knowledge of dystonia, a movement disorder that can affect people of all ages; this contributes to diagnostic delay and poor quality of life. We investigated whether a brief educational intervention could improve knowledge of dystonia amongst medical students. We conducted a systematic review on undergraduate knowledge of dystonia and created an eight-minute video on the condition. We invited medical students at the University of Leeds, UK, to answer 15 multiple choice questions before and immediately after watching the video, and again one month later. Only one previous study specifically assessed medical students’ knowledge of dystonia whilst five others tested their knowledge of movement disorders, or neurology generally, with some questions on dystonia. Of the University of Leeds medical students, 87 (100%), 77 (89%) and 40 (46%) completed the baseline, immediate-recall and delayed-recall questionnaires, respectively. The mean score for students who completed all three questionnaires increased from 7.7 (out of 15) to 12.5 on the immediate-recall questionnaire (*p* < 0.001), and to 10.1 on the delayed-recall questionnaire (*p* < 0.001). At baseline, 76% of students rated their confidence in recognising dystonia as low. After watching the video, 78% rated their confidence as a high, and none rated it low. A brief video improved their knowledge substantially, with sustained effects. This method could be incorporated into medical curricula to reduce diagnostic delays.

## 1. Introduction

Dystonia is a neurological condition characterised by abnormal muscle contractions which lead to atypical movements and/or postures, usually of a twisting nature [1]. It can affect all ages, including older adults. Prevalence is estimated to be approximately 16 per 100,000 persons [2]. Adequate management, using oral medications, botulinum toxin injections and physiotherapy, leads to symptom control and improved quality of life [3].

Dystonia may be primary/idiopathic or secondary to another neurological condition. It may affect a single body part (focal) or more than one body part (multi-focal). Focal dystonia has the potential to spread to contiguous body parts (segmental dystonia). The initial site of onset is related to the risk of spread. For example, dystonia of the peri-ocular muscles (blepharospasm) is more than twice as likely to spread compared to dystonia of the neck muscles (cervical dystonia) or hands [4].

In addition to motor symptoms, common non-motor symptoms include mood disorders, such as depression and anxiety, and pain [5,6]. Focal dystonia often exhibits task-specificity whereby involuntary muscle contractions only occur during specific physical tasks, usually of a repetitive nature; for example, writer’s cramp, golfers’ cramp and musician’s dystonia [7,8]. A ‘sensory trick’ (also called ‘geste antagoniste’), such as touching the face or head in a certain spot to temporarily relieves symptoms, is highly characteristic of dystonia, though the physiological mechanisms behind this are not completely understood [9,10]. Primary dystonia can usually be differentiated from functional dystonia through careful evaluation for the presence of positive clinical features of functional movement disorders, such as sudden onset and variability with distraction [11,12].

Focal onset dystonia in older adults may be one of the first clinical signs of a neurodegenerative condition, such as Parkinson’s disease (PD). Many people with dystonia experience diagnostic delays, inappropriate management and delayed referral to specialist services [13]. One study found that patients typically experienced a median delay of two years between symptom onset and diagnosis and saw three clinicians before diagnosis [14]. Delayed diagnosis is clinically important as it is associated with increased complications of untreated dystonia, such as contractures, side-effects from inappropriate treatments [14] and high rates of depression [15].

The ‘Dystonia Europe’ collaboration has found that healthcare professionals’ limited knowledge of dystonia is a major contributing factor to diagnostic delays [16]. One study found that only 10% of patients referred to a movement disorders clinic had been given a working diagnosis of dystonia in primary care and in approximately 1/3rd of cases, general practitioners had initially referred patients to a non-neurologist [17]. One method to reduce diagnostic delays is to improve the knowledge of dystonia among medical students. This may help all doctors in the future to recognise dystonia and refer appropriate patients promptly.

## 2. Materials and Methods

We conducted a systematic review of the literature and a prospective interventional cohort study.

### 2.1. Systematic Review

#### 2.1.1. Inclusion and Exclusion Criteria

Inclusion criteria comprised papers written in English, based on medical students’ knowledge and/or awareness of dystonia, which incorporated video/online education and had full-text availability. Papers were excluded if they did not meet the aforementioned criteria or were case reports, case series, review articles or abstracts.

#### 2.1.2. Scoping and Literature Search

Synonyms for ‘medical students’, ‘knowledge’ and ‘video-education’ were cross-referenced with ‘dystonia’ individually (Figure 1). The papers were then pooled to obtain the results of the scoping search. We modified the inclusion criteria, considering the results of the scoping search, to include abstracts and studies without video-education to yield a greater number of papers.

We searched the literature via three medical databases (PubMed, Embase, MEDLINE) and four grey literature databases (Semantic Scholar, WorldCat, OpenGrey and ProQuest), using the same strategy as depicted in Figure 1.

Duplicates were removed. Titles and abstracts were read to ensure papers were relevant. The remaining papers were read in their entirety (by SK) to ensure they met the inclusion criteria. A second opinion from coauthors was to be sought if there was doubt as to whether inclusion criteria were met, though this was unrequired.

### 2.2. Medical Student Cohort Study

#### 2.2.1. Brief Video Intervention

We created an eight-minute video featuring three patients with common forms of adult-onset dystonia: cervical (neck), upper limb (dystonic hand tremor and writer’s cramp) and oro-facial (oromandibular and blepharospasm). Patients were recruited from movement disorder clinics at Leeds Teaching Hospitals NHS Trust (LTHT) and had previously been diagnosed as having adult-onset dystonia by a neurologist who specialises in movement disorders (JA or JC). The final scripted video was reviewed for clinical accuracy by the medically-qualified authors (JA, JC).

We chose video-education intervention as it has been shown to improve knowledge retention [18] and reduce cognitive load through congruent presentation of visual and auditory information [19].

#### 2.2.2. Data Collection—Questionnaires

The ‘before and after’ questionnaire methodology is an established evaluation technique for technology-enhanced learning media [20,21]. Part one of the questionnaire comprised questions on demographic details, career aspirations, prior exposure to dystonia education and a Likert scale to indicate confidence in recognising dystonia. Part two consisted of 15 multiple choice questions (MCQs) about dystonia, including case-based scenarios that were chosen to mirror the format of undergraduate medical assessments within the UK (see Section A.1). We included three questions previously used by Jabir et al. [22] in their study of medical student awareness (definition, prevalence, inheritance) of dystonia, and modified a fourth by questioning how the prevalence of dystonia compares to PD rather than to Motor Neurone Disease. All questions were reviewed for clinical accuracy by JA and JC. The questionnaire was developed using Google Forms and data was automatically collected into a secure Microsoft Excel spreadsheet, which was subsequently downloaded for offline analysis.

#### 2.2.3. Recruitment

##### Medical Students

The project was advertised to medical students across all year groups at the University of Leeds (UoL), UK, on a regular basis over a twelve-month period throughout 2020–2021. Students were invited by an email sent to their UoL address, which contained a project information sheet and instructions on how to participate.

Students were asked to complete the baseline questionnaire (BQ). They were then asked to watch the video, which was uploaded onto the UoL virtual learning platform, and complete the immediate-recall questionnaire (IRQ). Finally, they were asked to complete the delayed-recall questionnaire (DRQ) one month later. The three questionnaires were identical in content and order of questions. Students were not given the correct answers until the end of the study. Consent was obtained by asking students to tick a box at the start of each questionnaire.

##### Patients

Patients attending the dystonia clinic at LTHT were invited to feature in the video. Those who expressed an interest were provided with a patient information sheet. They had the opportunity to ask questions and provided written consent on the day of their clinic appointment. All data handling adhered to General Data Protection Regulations and the Data Protection Act 2018.

#### 2.2.4. Ethical Approval

Ethical approval for medical student participation was obtained from the Head of the Bachelor of Medicine and Surgery (MBChB) programme at UoL. Exemption from School of Medicine (SoM) ethics review was granted by staff overseeing the Research, Evaluation and Special Studies module. The completion of the NHS Health Research Authority decision tool indicated that our project was exempt from the Integrated Research Application System (IRAS) ethics approval process, and this was confirmed by the LTHT IRAS sponsor representative.

#### 2.2.5. Statistical Analysis

We used a two-tailed, paired, independent samples *t*-test to calculate group differences between performance on the BQ and IRQ, and the BQ and DRQ. Only data for which pre- and post-intervention scores were available was used for the paired *t*-test. We used a two-tailed, unpaired, independent sample *t*-test (for unequal variances and sample size) to investigate whether performance on the BQ was associated with covariables such as receiving prior education on dystonia or seeing a patient with dystonia or observing dystonia being included in a differential diagnosis. We assessed associations between confidence in recognising dystonia and the aforementioned latter two independent variables using a Chi-squared test. Confidence scores on the 5-point Likert scale were divided into three groups before the Chi-squared tests were conducted: low (1 or 2), medium (3) and high (4 or 5). A single-factor between-groups Analysis of Variance (ANOVA) was used to investigate whether a higher year in medical school was associated with superior knowledge of dystonia. Homogeneity of variances was assessed via Levene’s test beforehand. Statistical significance was set at a *p* value of <0.05.

## 3. Results

### 3.1. Systematic Review

Figure 2 illustrates the process and outcomes of the systematic review.

The characteristics of the studies returned by the systematic review are shown in Table 1. Only one previous study specifically assessed medical student knowledge of dystonia [22], whilst three tested their knowledge of movement disorders generally, including dystonia [23,24,25], or their knowledge of general undergraduate neurology, including dystonia [26,27]. We did not identify any studies that used video-education to improve undergraduate knowledge of dystonia.

### 3.2. Cohort Study

Medical students from all years completed the questionnaire. Most respondents were undergraduate students (88.5%). Of the students, 3% expressed an interest in specialising in neurology.

87 students completed the BQ (5.8% response rate), 77 students (88.5% of the baseline participants) completed the IRQ, and 40 students (46%) completed the DRQ. The highest number of respondents were from year three (31%). The two most common age brackets of the medical students were 22–25 years and <22 years.

The mean score on the questionnaire was 7.7 out of 15 (range 3–12) on the BQ, 12.5 (9–15) on the IRQ and 10.0 (2–13) on the DRQ (Figure 3).

Based on the 77 students who completed both questionnaires, there was a significant improvement in knowledge of dystonia after the brief video intervention, with BQ scores increasing from a mean (SD) of 7.9 (2.1) to 12.5 (1.5) on the IRQ, (*p* < 0.001). Data for the 40 students who completed all three questionnaires showed scores of 7.7 (2.0) at baseline, 12.5 (1.3) on the IRQ (*p* < 0.001), and 10.1 (2.2) on the DRQ (*p* < 0.001)-suggesting a sustained improvement in knowledge.

For each individual MCQ, there was an increase in the number of students who provided a correct response between the BQ and the IRQ (Figure 4). The increase was particularly substantial for questions 4, 5, 8 and 10, which were based on the definition of Meige syndrome, a case-based scenario that required students to recognise ‘right torticollis’ (neck/cervical dystonia) from a description of it, recognising the commonest types of dystonia and the prevalence of dystonia in the UK, respectively.

The number of correct responses fell for all questions on the DRQ except for MCQs 1 and 12, which asked the definition of dystonia and whether it is possible to inherit dystonia respectively.

Progression through medical school, as measured by year group, was associated with a higher mean score on the BQ; the mean scores (SD) for year one to five were 6.7 (3.9), 7.5 (3.3), 7.7 (4.4), 8.5 (5.4) and 8.6 (4.1), respectively. However, this trend was not statistically significant (*p* = 0.09; Levene’s F test, *p* = 0.77).

At baseline, 40.2% of students had never heard of dystonia and 75.9% had never received any teaching on dystonia or seen a patient with dystonia. Students who had received prior education on dystonia performed significantly better (*p* < 0.001), with a mean (SD) score of 8.9 (4.0), compared to those who did not (mean score of 7.5 (4.1)). Similarly, students who had previously seen a patient with dystonia performed better (*p* < 0.001), with a mean score of 9.0 (3.6)), compared to those who had not (mean score of 7.4 (4.1)).

Before watching the video, 75.9% of students rated their confidence in recognising dystonia as low, and just 3.4% rated it as high. Immediately after watching the video, 78.3% rated their confidence as a high, and none rated it low (Figure 5); 72.5% of the students stated that they felt more confident in recognising cervical dystonia, whilst 47.5% felt more confident in recognising hand or facial dystonia.

## 4. Discussion

The short video-education intervention significantly improved knowledge of dystonia amongst medical students, and this was sustained one month later. Mean DRQ scores were lower than mean IRQ scores, which was expected as students typically perform less well with time; this is widely seen in pedagogical studies using pre- and post-intervention questionnaires to assess the impact of educational interventions, especially those using visual learning materials [28,29]. Knowledge of each MCQ, except the definition of dystonia and whether it is possible to inherit it, declined on the DRQ compared to the IRQ. However, performance on the DRQ was still superior.

Despite higher year groups achieving higher scores, mean BQ scores were not significantly different. This suggests that the next cohort of junior doctors (year five/final year medical students) are no more familiar with dystonia than first year medical students at UoL—highlighting a gap in training.

Most students in our study had never received prior teaching on dystonia or even heard of it, consistent with previous work [22]. The minority of students who had received prior teaching on dystonia, seen a patient with it or observed it being included in a differential diagnosis performed significantly better on the BQ and had greater confidence in recognising the condition. This finding supports our belief that a relatively short video-education intervention has the potential to reduce the rate of misdiagnosis of dystonia amongst future clinicians.

The two MCQs that showed the largest increase in knowledge were those asking about the commonest types of dystonia and knowledge of prevalence of dystonia in the UK (from 15.3% to 59.7% and 11.5% to 87% on the BQ and IRQ, respectively). Awareness of both facts is important for junior doctors and should enhance their ability to recognise dystonia.

Videos have previously been utilised to improve knowledge of neurological conditions such as PD and epilepsy [18], amongst others [26,30,31]. They are particularly valuable in teaching dystonia as they allow dynamic clinical signs such as twisting movements to be demonstrated. Practical benefits include remote use, which was advantageous for our study as it took place during the COVID-19 pandemic.

### Strengths and Limitations

This was the first study to investigate knowledge and awareness of dystonia using an online intervention and the second to specifically investigate medical student awareness of dystonia after Jabir et al. (2012) [22]. The use of pre- and post-video questionnaires provided objective data to assess the impact of the intervention [28,29]. The MCQ format allowed objective scoring of answers [32]. The questionnaires and video-education interventions were brief, which is likely to have been an important factor in motivating students to participate. All information included in the video and questionnaires was verified by the consultant neurologist co-authors (with expertise in dystonia) to ensure clinical accuracy. Its online nature makes it a permanent resource that is easy to share.

A significant weakness of the study is the low response rate and drop-out between all questionnaires, particularly the BQ/IRQ and the DRQ. There are several potential explanations for this. For example, some students may simply have forgotten to complete the DRQ. Medical students receive a high volume of invitations for research studies on a regular basis, which may result in them selecting the ones related to their career aspiration or those they find most interesting. The timing of the study could have reduced student retention because the DRQ coincided with UoL examinations.

The online platforms that we utilised to conduct our study did not allow us to record the number of times each student watched the video, which potentially offered students a chance to check their answers to the MCQs. Similarly, it is possible that students used resources other than the video to answer the questionnaires. This form of academic dishonesty is a widely accepted challenge within remote assessments [33,34].

## 5. Conclusions

Our study showed that an eight-minute video-education intervention significantly improved UoL SoM undergraduate knowledge of dystonia and confidence in recognising it. Video-education is an efficient, low-cost intervention that can easily be incorporated into undergraduate medical curricula, which could be useful in boosting knowledge of dystonia amongst the ‘doctors of tomorrow’.

## Figures and Tables

**Figure 1 geriatrics-07-00058-f001:**
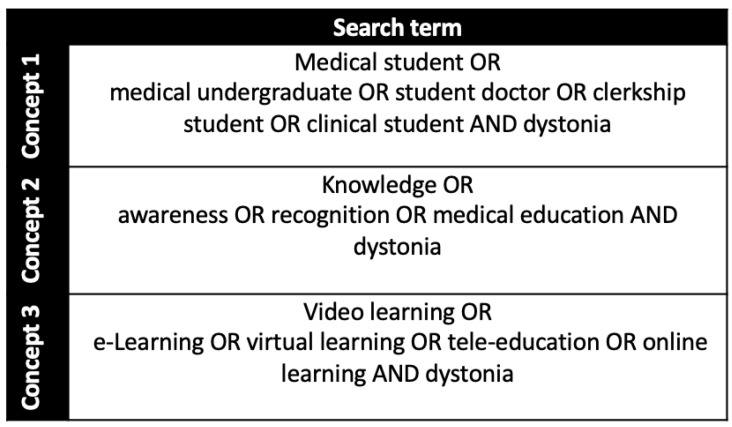
Terms used in scoping search.

**Figure 2 geriatrics-07-00058-f002:**
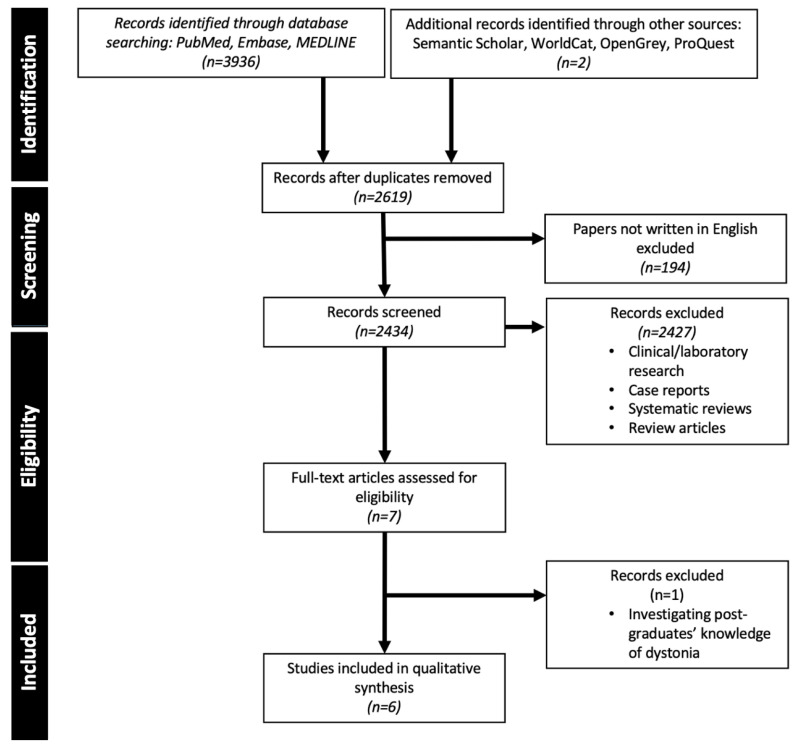
Preferred Reporting Items for Systematic reviews and Meta-Analyses (PRISMA) flow diagram illustrating literature search strategy.

**Figure 3 geriatrics-07-00058-f003:**
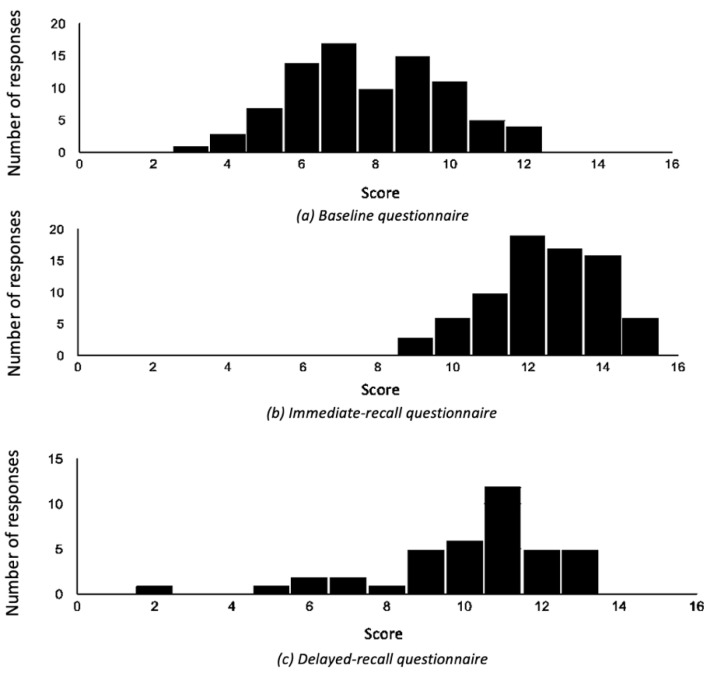
Medical student scores on (**a**) baseline, (**b**) immediate- and (**c**) delayed-recall questionnaires.

**Figure 4 geriatrics-07-00058-f004:**
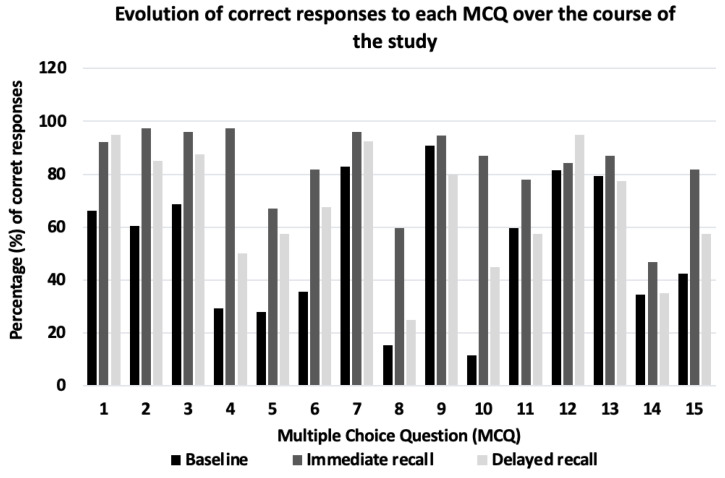
Percentage (%) of correct responses to each MCQ (1–15) over course of study.

**Figure 5 geriatrics-07-00058-f005:**
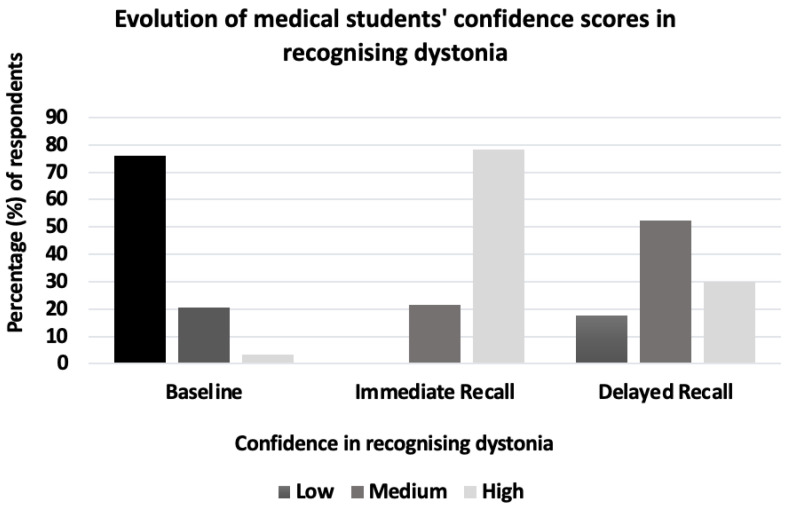
Medical student confidence in recognising dystonia on baseline, immediate- and delayed-recall questionnaire. Prior teaching on dystonia was associated with significantly greater confidence in recognising it in a patient (*p* < 0.001), as was seeing a patient with dystonia or observing it being included in a differential diagnosis (*p* < 0.001).

**Table 1 geriatrics-07-00058-t001:** Characteristics of the studies included within the systematic review.

Authors	Location	Number of: Universities; Students	Medical Student Year Group	Was the Study Specifically Based on Awareness of Dystonia or Movement Disorders Generally?	What Intervention was Used?
Dominguez et al.,2018 [26]	USA	1; 135	Year 3	Undergraduate neurology (including movement disorders)	Video lectures; did not include patient videos
Jabir et al., 2012 [22]	UK	1; 51	Year 3, 4, 5	Dystonia	None
Lawal et al., 2012 [23]	Nigeria	1; 228	Year 4 and 5	Movement disorders	None
Cubo et al., 2017 [24]	Argentina and Cameroon	5; 151	Final year	Movement disorders	Videoconferences including patient videos
Nwazor and Okeafor, 2019 [25]	Nigeria	1; 79	Final year	Movement disorders	None
Menkes and Reed, 2008 [27]	USA	1; 415	Year 3 and 4	Undergraduate neurology (including movement disorders)	Didactic teaching sessions

## Data Availability

Not applicable.

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
