# Peer review of "Increased Knowledge of Adult-Onset Dystonia Amongst Medical Students via Brief Video Education: A Systematic Review and Cohort Study"

_geriatrics, 2022, doi:10.3390/geriatrics7030058_

Round 1

Reviewer 1 Report

The authors reported a study on educational knowledge of dystonia. The paper is well-written, and the findings are interesting. I have some comments to the authors:

1) I would ask the authors to add a new paragraph in the introduction describing some general, but very important, features of dystonia. I do think that this new paragraph could help the readers (especially if not neurologist) to better understand the topic and it’s certainly in line with the main aim of the study. In particular the authors should cover these points:

i) Sensory trick - brief description of the phenomenon. The authors should add these references to help the reader to better understand [Dagostino S. et al. Sensory trick in upper limb dystonia. Park Relat Disord 2019; V.F. Ramos, B.I. Karp, M. Hallett. Tricks in dystonia: ordering the complexity, J. Neurol. Neurosurg. Psychiatry 2014]

ii) Non-motor symptoms in dystonia - a general description of the main symptoms would be enough, with particular references to psychiatric disorders and pain. Here some references to be added [Tinazzi et al. Demographic and clinical determinants of neck pain in idiopathic cervical dystonia. Journal of Neural Transmission 2020; Kuyper DJ, et al. Nonmotor manifestations of dystonia: a systematic review. Mov Disord. 2011]

iii) Task-specificity - this is a very peculiar characteristic of some type of dystonia, so a general comment describing the semeiology and the differences with non-task specific dystonia would be beneficial. I recommend these important references [Torres-Russotto D, et al. Task-specific dystonias: a review. Ann N Y Acad Sci 2008; Defazio et al. Idiopathic Non‐task‐Specific Upper Limb Dystonia, a Neglected Form of Dystonia. Movement Disorders 2020].

iv) A brief mention of the phenomenon of dystonia spread, stressing the possibility that the disorder can affect other body part during the disease course. This recent work from Dystonia Coalition would be very helpful [Berman B et al. Risk of spread in adult-onset isolated focal dystonia: A prospective international cohort study. JNNP 2019]

v) The difficult differential diagnosis with functional movement disorders is a very challenging topic in Neurology, and particularly with dystonia. In my view it’s important that the reader, especially if not neurologist, can immediately understand the main differences between the two condition. These two milestones in the field should be added in the text. [Lidstone S. et al. Functional movement disorder gender, age and phenotype study: a systematic review and individual patient meta-analysis of 4905 cases. JNNP 2022; Frucht L. et al. Functional Dystonia: Differentiation From Primary Dystonia and Multidisciplinary Treatments. Frontiers in Neurology 2021.]

2) Please include also part one of the questionnaire in the Appendix.

Author Response

1) Thank you for pointing this out and for giving us such detailed feedback for this - your review has facilitated us significantly improving the article. We have added new sections to the introduction as suggested, and agree this gives greater clarity and context for the reader 2) Many thanks for pointing this out. Part 1 of the questionnaire has now been added too 

Reviewer 2 Report

In this paper, the authors present the results of a study on the effect of educational video on adult-onset dystonia knowledge in medical students. The paper is well written and the study is sound. I have only a few suggestions to further improve the study.

The title of the article is misleading. Before reading the article, I thought the article was about the new findings on dystonia in adults. I would suggest changing the title to one that reflects the content of the article. E.g., "Increased knowledge in medical students about adult dystonia through the use of videos...". This should then be explained in the "Discussion" section of the article-particularly noting that the use of videos in education when dealing with movement disorders is of paramount importance!

If possible, it would be nice if you could attach the video to the article. This would increase the quality of the article immensely.

Author Response

  1. Thank you for this helpful suggestion. We have changed the title as suggested to " Increased knowledge of adult-onset dystonia amongst medical students via brief video education: A systematic review and cohort study"
  2. In the methods section, we outlined "We chose video-education intervention as it has been shown to improve knowledge retention[9] and reduce cognitive load through congruent presentation of visual and auditory information[10]. Videos have previously been utilised to improve knowledge of neurological conditions such as Parkinson's disease (PD) and epilepsy[9], amongst others[11,12,13]. They are particularly valuable in teaching dystonia as they allow dynamic clinical signs such as twisting movements to be demonstrated. Practical benefits include remote use, which was advantageous for our study which took place during the COVID-19 pandemic". Based on your suggestion, we have now split this information so that the following expanded section is placed in the discussion section rather than methods (with updated references, as suggested by another reviewer): Videos have previously been utilised to improve knowledge of neurological conditions such as PD and epilepsy[18], amongst others[26,30,31]. They are particularly valuable in teaching dystonia as they allow dynamic clinical signs such as twisting movements to be demonstrated. Practical benefits include remote use, which was advantageous for our study which took place during the COVID-19 pandemic

3. Many thanks for your suggestion on including the video. We agree completely, and are in the process of confirming that all patients who featured in the video are still happy for us to disseminate/publish it